# Charge transfer to ground-state ions produces free electrons

D. You[1,2], H. Fukuzawa[1,2], Y. Sakakibara[1,2], T. Takanashi[1,2], Y. Ito[1,2], G.G. Maliyar[1,2], K. Motomura[1,2], K. Nagaya[2,3], T. Nishiyama[2,3], K. Asa[2,3], Y. Sato[2,3], N. Saito[2,4], M. Oura[2], M. Schöffler[2,5], G. Kastirke[5], U. Hergenhahn[6,7], V. Stumpf[8], K. Gokhberg[8], A.I. Kuleff[8], L.S. Cederbaum[8] & K. Ueda[1,2]

Inner-shell ionization of an isolated atom typically leads to Auger decay. In an environment, for example, a liquid or a van der Waals bonded system, this process will be modified, and becomes part of a complex cascade of relaxation steps. Understanding these steps is important, as they determine the production of slow electrons and singly charged radicals, the most abundant products in radiation chemistry. In this communication, we present experimental evidence for a so-far unobserved, but potentially very important step in such relaxation cascades: Multiply charged ionic states after Auger decay may partially be neutralized by electron transfer, simultaneously evoking the creation of a low-energy free electron (electron transfer-mediated decay). This process is effective even after Auger decay into the dicationic ground state. In our experiment, we observe the decay of $Ne^{2+}$ produced after Ne 1$s$ photoionization in Ne–Kr mixed clusters.

[1] Institute of Multidisciplinary Research for Advanced Materials, Tohoku University, Sendai 980-8577, Japan. [2] RIKEN SPring-8 Center, Kouto 1-1-1, Sayo, Hyogo 679-5148, Japan. [3] Department of Physics, Kyoto University, Kyoto 606-8502, Japan. [4] National Metrology Institute of Japan, AIST, Tsukuba 305-8568, Japan. [5] Institute for Nuclear Physics, Johann Wolfgang Goethe University Frankfurt, Frankfurt 60438, Germany. [6] Leibniz Institute of Surface Modification, Leipzig 04318, Germany. [7] Max-Planck-Institute for Plasma Physics, Greifswald 17491, Germany. [8] Theoretische Chemie, Physikalisch-Chemisches Institut, Universität Heidelberg, Heidelberg 69120, Germany. Correspondence and requests for materials should be addressed to K.U. (email: ueda@tagen.tohoku.ac.jp).

When a light atom is irradiated by X-rays, the most likely process is photoionization of an inner-shell electron, followed by the emission of one or several Auger electrons of relatively high energy (>200 eV; ref. 1). When condensed matter is irradiated by X-rays however, the majority of electrons emitted into vacuum is of low energy (<20 eV). The process of low-energy electron production by X-rays is therefore highly indirect. These low-energy electrons are traditionally believed to be secondary electrons produced via inelastic scattering of a photoelectron or Auger electrons by the surroundings. In a biological system, a high potency for genotoxic effects has been assigned to these low energy electrons[2,3].

In 1997, Cederbaum et al.[4] predicted another pathway to produce a low-energy electron in a loosely bound system: If an ion in such system is produced in an excited state, it may transit to a lower electronic state, while simultaneously a low-energy electron is emitted from a neighbouring atom or molecule. This interatomic/intermolecular Coulombic decay process (ICD) has since been the topic of both theoretical[5] and experimental[6] studies, and it was shown to be an important source of slow electrons in water[7,8]. In particular, excited ionic states populated in an Auger process were shown to decay by ICD in an environment[9–11]. As both slow electrons and reactive radicals are produced in such Auger-ICD cascades, their relevance for radiation damage and radiation therapy was suggested[12,13].

In ICD, the decay starts from an electronically excited ionic state of the atom or molecule that was originally ionized. The majority of Auger transitions, however, populate the ionic ground state, or states that are weakly excited. Although these states cannot decay by ICD, surprisingly Stumpf et al.[14] predicted in 2013 that they still may decay electronically in a heterogeneous system. Here, another interatomic process called electron transfer mediated decay (ETMD)[15] comes into play, in which 'electron transfer' to an ion is accompanied by electron emission from the donor, or from a second neighbour (Fig. 1). According to the number of sites involved, the reaction is classified as ETMD(2) or ETMD(3). Both variants were demonstrated experimentally[16,17], but ETMD was seen as a minor decay channel since in systems considered initially it could not compete with ICD[15]. It can become the dominant relaxation pathway, however, for configurations in which ICD is energetically forbidden[17]. As an example, Stumpf et al. presented ab initio calculations for the NeKr$_2$ trimer and showed that ETMD(3) takes place between Ne$^{2+}$ in its ground state configuration ($2p^{-2}$), and the Kr neighbours (Fig. 1). In addition to emitting a slow electron, ETMD leads to the partial neutralization of Ne$^{2+}$ and the double ionization of its environment, resulting in Ne$^+$ and two Kr$^+$ ions. According to ref. 14, the process takes place within a few picoseconds in a trimer, which could shorten to below 1 ps owing to environmental effects in larger systems. Such ETMD-driven neutralization has a broad significance and was predicted to play an important role in radiation damage to biomolecules subjected to the action of ionizing radiation[18]. Here, we show experimental evidence for this efficient neutralization process using larger Ne–Kr mixed clusters as a prototype example.

## Results

**Electron transfer mediated decay in NeKr clusters.** Figure 1 sketches the series of events from the initial Ne 1s photoioniza-tion to Coulomb explosion following ETMD(3) in Ne–Kr mixed clusters. We expect the production of three singly charged ions (Ne$^+$, 2Kr$^+$) and three electrons (photoelectron, Auger electron and ETMD electron). To search for the experimental signature of

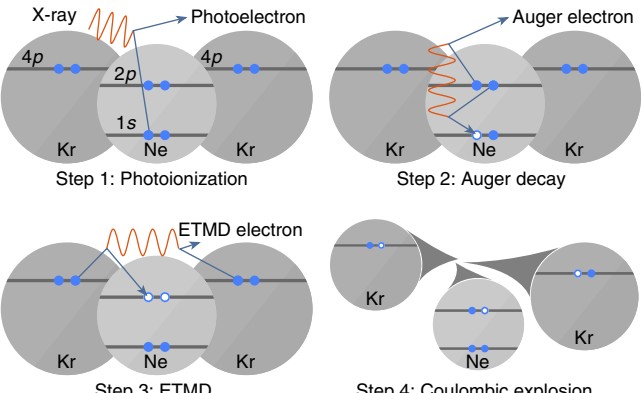

**Figure 1 | Process investigated.** First step (photoionization): an ionizing X-ray photon ejects a Ne 1s electron (photoelectron) from a Ne atom in the Ne–Kr mixed cluster. Second step (Auger effect): Auger decay of the resulting Ne$^+$ ion leads to Ne$^{2+}$ with two holes in the valence shell and an ejected electron (Auger electron). Third step (ETMD(3)): one of the electrons of a neighbouring Kr atom fills one of the Ne valence holes and one of the valence electrons of another Kr atom is ejected (ETMD electron). Fourth step (Coulomb explosion): the cluster explodes by Coulomb repulsive forces and releases one Ne$^+$ and two Kr$^+$ ions. Active electrons are indicated by filled blue discs and the positive charges created by empty blue circles.

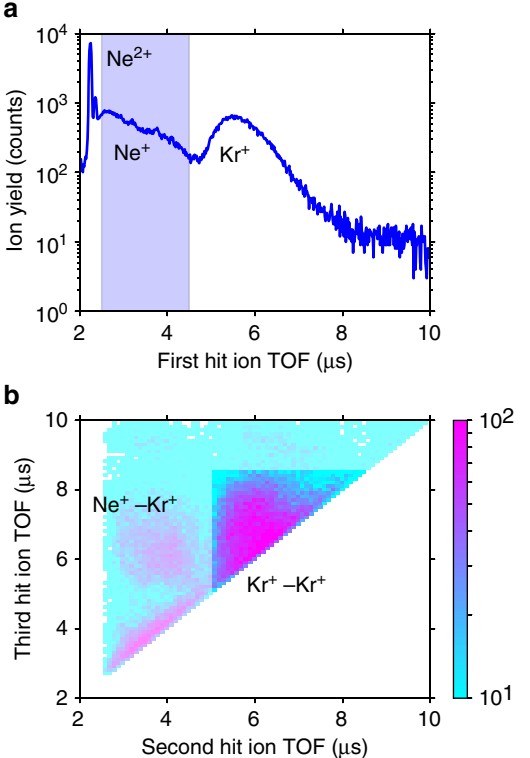

**Figure 2 | Time-of-flight spectra of ions released from Ne–Kr mixed cluster.** Results filtered for events in which three ions were detected in coincidence. The photon energy used was 888 eV. (**a**) Time of flight (TOF) of the first ion arriving at the detector. The shaded region was used to select events, in which a Ne$^+$ ion was detected. (**b**) TOFs of the second versus the third ion, detected in coincidence with Ne$^+$ as the first ion. Scale bar indicates counts.

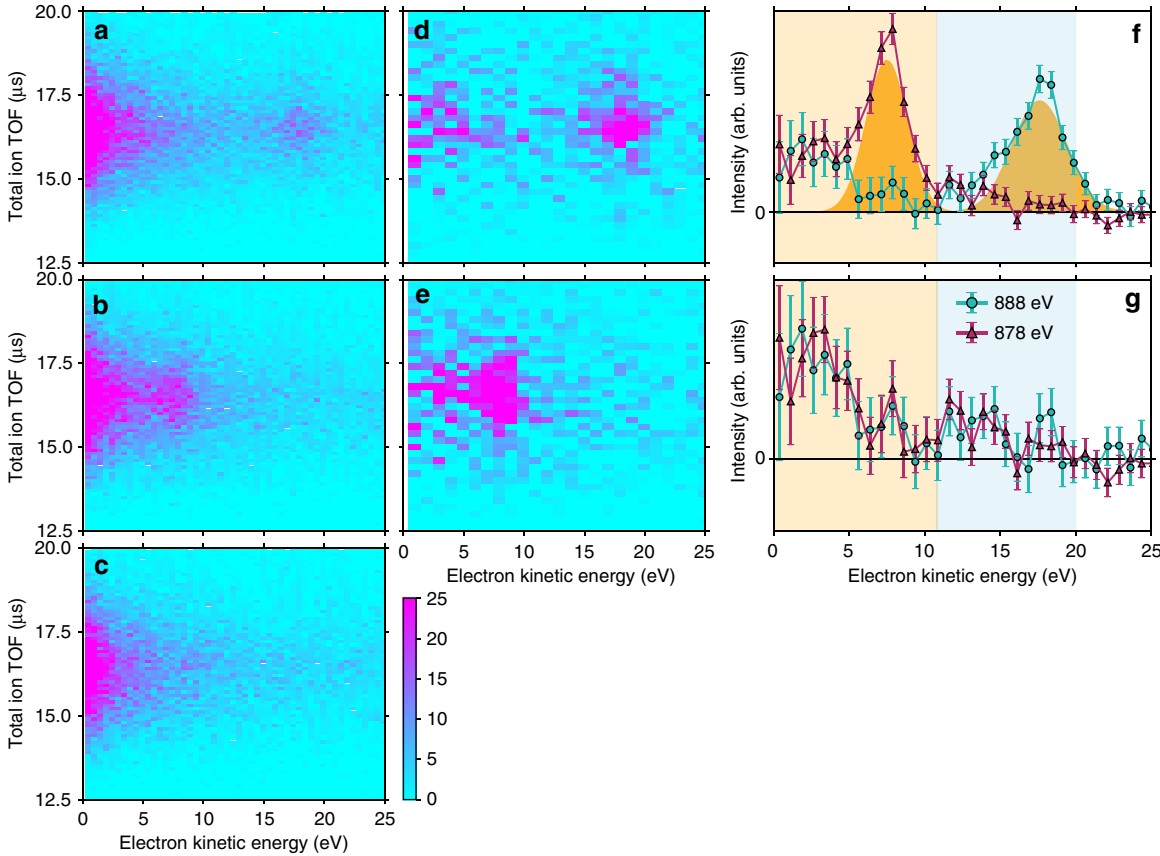

**Figure 3 | Electrons measured in coincidence with the target ions.** Kinetic energy of electrons detected in coincidence with three ions, for total ion TOFs pertaining to one Ne$^+$ and two Kr$^+$ ions. Panels correspond to (**a**) a photon energy of 888 eV, (**b**) 878 eV and (**c**) of 860 eV (below the Ne 1s ionization threshold). In total, in **a**–**c**, 18,616; 17,610; and 13,086 events are plotted, respectively. Correlation map of an electron and these three ions after subtracting the contributions from Kr ionization (**c**), for a photon energy of (**d**) 888 eV and (**e**) 878 eV. Scale bar indicates counts in **a**–**e**. (**f**) Projections of **d** and **e** on the electron energy axis. Two Gaussian functions fitted to the photoelectron peaks are also shown as yellow fill patterns. (**g**) The same, after subtracting the contributions from photoelectrons, fitted by two Gaussian functions. Error bars in **f** and **g** are defined as standard deviation. Electron spectra in coincidence with the target ions for energy range between 0 and 40 eV before subtraction are shown in Supplementary Fig. 3.

these particles, we irradiated free Ne–Kr mixed clusters by X-rays at photon energies of 878 and 888 eV, that is, 8 and 18 eV above the Ne 1s ionization threshold. For every absorbed photon, the momenta of the electrons and ions produced were measured in coincidence. In the following, we will discuss these results.

**Ion time-of-flight spectra.** First, we extracted all events in which the above three ions were produced (see Supplementary Note 1 for details). Figure 2 shows time-of-flight (TOF) spectra of ions released after absorption of a photon of 888 eV energy. Different mass-to-charge ratios give rise to peaks at different TOF values. The peaks are broadened by the initial velocities of the ions. We expect Ne$^+$ and Kr$^+$ to appear around 3.2 and 6.5 μs, respectively, and select only events in which we detected a Ne$^+$ ion, followed by two other ions. Ne$^+$, followed by Kr$^+$–Kr$^+$, is the most abundant such triple (Fig. 2b). The total TOF of all three ions can be used to distinguish different ion combinations, and amounts to approximately 16 μs for Ne$^+$ + 2Kr$^+$. More information is contained in the ion spectra plotted versus ion kinetic energy in Supplementary Figs 1 and 2, and discussed in Supplementary Note 2.

**Electron energy spectra.** We were able to go a decisive step further by measuring the correlation between ions and electrons. Figure 3a,b show the electron kinetic energy versus

total ion TOF for the events selected above. Regions at ∼18 eV kinetic energy in Fig. 3a and at ∼8 eV in Fig. 3b correspond to Ne 1s photoelectrons. This provides evidence that the ion triples are indeed produced after Ne 1s photoionization. In addition to the photoelectrons, significantly enhanced intensity is seen around 0–5 eV in both panels of Fig. 3a,b. Although we expect the ETMD electrons to appear in this energy region, we have to consider also other events triggered by Kr photoionization, since the photoabsorption cross-section of Ne is ∼60% of that of Kr at 900 eV (ref. 19).

To assess the contributions from Kr photoionization, we repeated the experiment at a photon energy of 860 eV (below the Ne 1s ionization threshold). At this energy, the photoabsorption cross-section of Ne is only ∼4% of that of Kr (ref. 19), and here we may attribute most events to Kr photoionization. Figure 3c depicts the correlation map measured at 860 eV. The intensity around 0–5 eV is weaker than in Fig. 3a,b. Panels (a) and (b) are scaled such that their intensity in the high energy background region between 25 and 40 eV in (a) and (b) agrees with panel (c) (see Supplementary Fig. 3 and Supplementary Note 3). Islands at 0–5 eV in Fig. 3a,b thus include contributions from both Ne and Kr photoionization. We may use Fig. 3c to subtract the contributions of Kr photoionization in the former panels, with results shown in Fig. 3d,e for 888 and 878 eV, respectively. After subtraction of the intensity from Kr photo-ionization, we may conclude that there are electrons at least in the

**Table 1 | Relative intensities of the various groups of electrons.**

| Electron energy (eV) | Assignment | Relative intensity | |
|---|---|---|---|
| | | Measured | Estimated |
| 8 or 18 | Photoelectron | 1.00 | 1.000 |
| 0–11 | ETMD, ICD (low) | 0.73 ± 0.09 | 1.000 <br> 0.163 |
| 11–20 | ICD (high) | 0.25 ± 0.05 | 0.194 |

Measured values are extracted from Fig. 3. For comparison, estimated values are shown, which are extracted from atomic Auger ratios and ICD branching ratios in small clusters. Depending on the photon energy used to ionize the Ne 1s shell in the cluster, the energy of the photoelectrons is around 8 or 18 eV. Experimental relative intensities for the ETMD and ICD contributions are valid for both photon energies. The estimated relative intensity of ETMD is 1.0 because all cascades end with ETMD, see Fig. 4.

range about 0–11 eV in addition to the Ne 1s photoelectrons. Our choice of the scaling factor in the background subtraction procedure described here is conservative, and its influence on our results is discussed in depth in Supplementary Note 4, with Supplementary Figs 4–7 and Supplementary Tables 1–4.

Finally, we determined the electron energy spectra shown in Fig. 3f by projecting Fig. 3d,e on the electron energy axis. We normalized the spectra to unit area of the intensity between 0 and 25 eV. In addition to contributions from photoelectrons, there are two components which do not shift with photon energy, as expected for ETMD. One appears at about 0–11 eV as already seen in Fig. 3d,e, and the other at about 11–20 eV. After subtracting the contributions from photoelectrons (Fig. 3g), we can see more clearly that the two low-energy components appear at both photon energies. The relative intensities of the components making up the electron spectra are summarized again in Table 1. The intensity of the 0–11 eV electrons is ~70% of that of the photoelectrons. This implies that when a Ne 1s photoionization event at the Ne–Kr interface occurs, almost always an electron with a low kinetic energy of several electronvolts is emitted from the Ne–Kr mixed cluster. A contribution of electron scattered inelastically to the low kinetic energy fraction cannot be ruled out in principle, but seems negligible in this experiment as discussed in Supplementary Note 5.

## Discussion

We now discuss in detail how these extra electrons are produced in the Ne–Kr mixed clusters (Fig. 4). We assume that branching ratios for Auger decay in atoms apply to our weakly bonded clusters as well[20]. After Ne 1s ionization, 93% of the resulting $Ne^+(1s^{-1})$ vacancies undergo Auger decay to $Ne^{2+}$ (ref. 21). Figure 4 depicts schematically the decay pathways via these dicationic states. When $Ne^{2+}(2p^{-2}\,^1D)$ and $Ne^{2+}(2p^{-2}\,^1S)$ are produced by Auger decay in the mixed clusters, they decay via ETMD(3) giving rise to $Ne^+$ and two $Kr^+$ and to ETMD electrons with energies of 0–4.5 eV ($Ne^{2+}\,(2p^{-2}\,^1D)$) and 1–8.5 eV ($Ne^{2+}\,(2p^{-2}\,^1S)$)[14]. When higher electronic states of $Ne^{2+}$ are produced by Auger decay, they continue to decay by ICD. $Ne^{2+}(2s^{-1}2p^{-1}\,^3P)$ decays into $Ne^{2+}(2p^{-2}\,^3P)$ emitting ICD electrons of 0–3.5 eV; the next state of higher energy, $Ne^{2+}(2s^{-1}2p^{-1}\,^1P)$, decays into $Ne^{2+}(2p^{-2}\,^1D)$ and $Ne^{2+}(2p^{-2}\,^1S)$ with a branching ratio of 5:1 (ref. 22) and emits ICD electrons of 8–11 eV and 5–7.5 eV, respectively. The highest excited state in the Auger spectrum, $Ne^{2+}(2s^{-2}\,^1S)$, first decays by ICD to $Ne^{2+}(2s^{-1}2p^{-1}\,^1P)$ emitting 0–2 eV electrons, which then decays by another ICD step, as described above. The last step after these Auger-ICD cascades, for 'all' states produced, is the decay by ETMD, as seen at the bottom of Fig. 4. Interestingly, also the dicationic state $Ne^{2+}(2p^{-2}\,^3P)$, which is

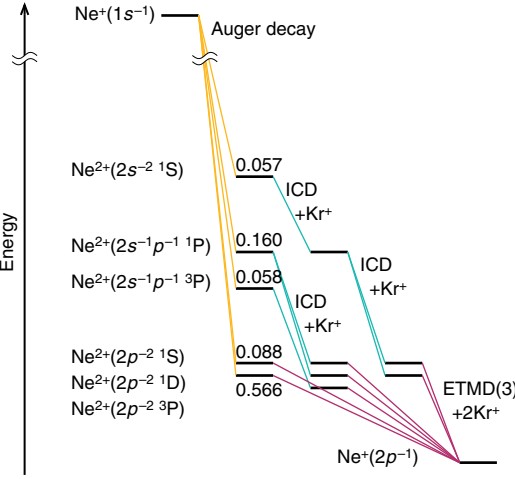

**Figure 4 | Schematic view of the decay pathways of Ne⁺ produced by Ne 1s photoionization in Ne–Kr mixed clusters.** The first step consists of Auger decay (indicated by yellow lines) of the $Ne^+(1s^{-1})$ ion, giving rise to various dicationic $Ne^{2+}$ states shown on the left, in the order of ascending energy. For each state, the abundance[25] is shown at the respective energy level. States with the electronic configuration $Ne^{2+}(2p^{-2})$, which are the most abundant, can only decay by ETMD(3), thus neutralizing $Ne^{2+}$ to $Ne^+$. This is indicated by magenta lines. The higher lying dicationic states produced by Auger decay first decay by ICD (indicated by turquoise lines) and then further by ETMD(3).

not populated by Auger is populated via ICD. The latter emits ETMD electrons in the range 0–1.5 eV. The above energies of the ICD and ETMD electrons were estimated from NeKr and NeKr₂, respectively. In the larger mixed clusters measured here, the electron energies may, of course, shift somewhat. Qualitatively similar, but more involved decay cascades are also expected after double Auger decay to $Ne^{3+}$ (6% branching ratio[21]).

In the electron emission events described above, the ICD process from $Ne^{2+}(2s^{-1}2p^{-1}\,^1P)$ to $Ne^{2+}(2p^{-2}\,^1D)$ leads to electrons of rather high kinetic energy, which can be identified with the intensity in the 11–20 eV range shown in Fig. 3f and assigned 'ICD (high)' in Table 1. All other ETMD and ICD electrons correspond to the 0–11 eV electrons, assigned 'ETMD and ICD (low)' in Table 1. In addition to the experimental results in Table 1, we also show estimated values of the relative intensities of the various groups of electrons by using the atomic Auger ratios and the branching ratio of the ICD from $Ne^{2+}(2s^{-1}2p^{-1}\,^1P)$. The estimated and experimental values are in satisfactory agreement, in view of the complexity of the involved multi-step ICD and ETMD processes.

Our interpretation is additionally supported by an analysis of the ion kinetic energy release (KER), shown in the Supplementary Fig. 2 and Supplementary Note 2. The KER spectrum clearly shows a component related to Ne photoionization, the energies of which are in reasonable agreement with a Coulomb explosion of $Ne^+$ and $2Kr^+$, in which some momentum is taken away by neutral fragments.

In summary, we have shown that the dicationic states produced via Auger decay following Ne $1s$ photoionization of Ne–Kr mixed clusters are subject to ETMD(3). The most abundant states in the Auger spectrum are at low energy and undergo ETMD directly. The dicationic states at higher energy undergo a multi-step decay, at which ETMD(3) occurs at the end of the cascade (Fig. 4). The ETMD step partially neutralizes the Ne dication and gives rise to the emission of low-energy electrons.

Although our experiment has provided evidence for the existence and the efficiency of Auger-ETMD cascade processes, the statistics of our data were not sufficient to assess it in all quantitative detail. More sophisticated modelling of the cascade processes as well as extended measurements should further improve our understanding.

Our results apply to relaxation pathways after interaction with energetic particles in a broad range of weakly bonded systems, for example, aqueous solutions. We show that non-local autoionization may occur even from states with no or a small amount of electronic excitation energy. Taking these processes into account is very important for understanding the chemical effect of radiation on a microscopic level.

## Methods

**Experiment.** The experiment was carried out at the b-branch of the beamline BL17SU of SPring-8 (ref. 23). The storage ring was operated in several-bunches mode providing 12 single bunches (1/14 filling bunches) separated by 342 ns. Circular polarization of the incident light was chosen to maximize the photon flux at the used photon energies (860, 878 and 888 eV; ref. 23).

The Ne–Kr mixed cluster beam was prepared by expanding mixed gas through an 80 μm nozzle at a stagnation pressure of 0.6 MPa. The molar mixing ratio of Ne : Kr was 60:1. Temperature of the nozzle was 160 K. The average cluster size for pure Ne clusters under these conditions is $\sim 10$ (ref. 24). Since Kr is condensed easier than Ne, the ratio of Kr to Ne in the clusters is larger than in the gas mixture. We estimated the ratio of Ne : Kr in the clusters as 4:1 from coincidence counts of the $Ne^+$–$Ne^+$–$Kr^+$ and $Ne^+$–$Kr^+$–$Kr^+$ ion sets, respectively, with Ne $1s$ photoelectrons, after subtracting contributions from Kr photoionization. For the estimate, we consider the decay of the $Ne^{2+}(2s^{-1}2p^{-1}\,^1P)$ excited dicationic state. This state will decay via ICD, for which two alternative routes exist however: The decay may involve either a neighbouring Kr atom ($Ne^{2+}$–Kr), or it may decay via ICD with a Ne atom ($Ne^{2+}$–Ne; ref. 20). The latter decay will lead to production of a $Ne^+$–$Ne^+$–$Kr^+$ ion triple, and is the only important channel giving that signature. The alternative route involving a Kr neighbour, like all other decay pathways considered here, will produce a $Ne^+$–$Kr^+$–$Kr^+$ signature. We thus multiply the abundance of $Ne^+$–$Kr^+$–$Kr^+$ with the branching ratio into the $Ne^{2+}(2s^{-1}2p^{-1}\,^1P)$ state, which is 17.2% (ref. 25) and relate it to the abundance of $Ne^+$–$Ne^+$–$Kr^+$ to arrive at the above result.

To measure ions and electrons produced by the series of events, we used an electron–ion three-dimensional momentum coincidence spectrometer[26–28]. The spectrometer consists of two TOF spectrometers equipped with delay-line type position-sensitive detectors. One of them detects ions and the other detects electrons. Those face each other with the reaction point between them. The lengths of the acceleration region and the drift region of the electron spectrometer are 33.0 and 67.4 mm, respectively. For the ion spectrometer, there are two acceleration regions and no drift region. The length of the first acceleration region is 16.5 mm and that of the second one is 82.5 mm. The TOF spectrometer for electrons is equipped with a hexagonal multi-hit position-sensitive delay-line detector of effective diameter of 120 mm, while that for the ions is of effective diameter of 80 mm. In the present experiments, the static extraction field was set to 1.7 V mm$^{-1}$, and that of the second acceleration region for the ions was set to 21 V mm$^{-1}$. A uniform magnetic field of 6.8 G was superimposed to the spectrometer by a set of Helmholtz coils outside the vacuum chamber. The knowledge of position and arrival time on the particle detectors allows us to extract information about the three-dimensional momentum of each particle. Approximately 18,000 ion–ion–ion–electron coincidences were collected over about 4.5 h acquisition time in each of Fig. 3a,b.

**Data availability.** All relevant data are available from the corresponding author on request.

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

## Acknowledgements

We are grateful to late Professor Makoto Yao for an invaluable contribution. The experiment was performed at BL17SU in SPring-8 with the approval of RIKEN (Proposal No. 20150061). This study was supported by the X-ray Free Electron Laser Priority Strategy Program of the Ministry of Education, Culture, Sports, Science and

Technology of Japan (MEXT), by the Japan Society for the Promotion of Science (JSPS) KAKENHI Grant Number 15K17487, by the Cooperative Research Program of Network Joint Research Center for Materials and Devices: Dynamic Alliance for Open Innovation Bridging Human, Environment and Materials, and by the IMRAM project. G.K. and M.S. acknowledge financial support by the Federal Ministry of Education and Gesearch (BMBF). V.S., K.G. and U.H. gratefully acknowledge the financial support of the Deutsche Forschungsgemeinschaft (Research Unit 1789). L.S.C. gratefully acknowledges the funding from the European Research Council/ERC Advanced Investigator Grant No. 692657. We thank C. Richter for a careful reading of the manuscript.

## Author contributions

K.U. conceived the research. D.Y., Y.S. and T.T. prepared the cluster beam. D.Y., H.F., Y.S., T.T., Y.I., G.G.M., K.M., K.N., T.N., K.A., Y.S., N.S., M.O., M.S. and G.K. performed the experiment. D.Y. analysed the data. D.Y., H.F. and K.U. prepared the manuscript with extensive suggestions from U.H., V.S., K.G., A.I.K and L.S.C. and contributions from all the other authors.

## Additional information

**Competing financial interests:** The authors declare no competing financial interests.

**Publisher's note**: 

