## [Peer Review File · Nature Communications]

Reviewers' comments:

Reviewer #1 (Remarks to the Author):

The manuscript deals with electronic relaxation pathways of a core ionized NeKr₂ cluster thanks to a collaboration between theoreticians and experimentalist. On one side, the state of the art theory is able to calculate ETMD processes which are, so far, been very difficult to experimentally observe.

The authors were able to extract coincidences between three ions : Ne⁺ and two Kr⁺ ions. From the experimental point of view, it would have been much better to learn about the Kinetic Energy Released of the dissociative NeKr₂ triply charged ion. Does the KER correspond to what can be expected from a simple Coulomb explosion picture ?

The data treatment (subtraction of below Neon 1s threshold) and comparison with above Neon threshold measurements suffers from the photoelectron peak which stands up in the middle of the spectrum and with low statistics. It would be nicer to appreciate the number of events measured to appreciate the statistical uncertainty. The figure 3c (measurements below the Ne 1s threshold) exhibits almost the same 2D spectrum when measured 8 or 18 eV above the Ne 1s threshold. It is then hard to believe that the authors well measured ETMD above the threshold.

I also have some doubt about the relative cross section of the ETMD. The authors claim that 70% of the 1s ionization event leads to ETMD. I wonder if the authors did take into account some events leading to, for instance Ne⁺ + Kr⁺ + neutral Kr or to Ne⁺⁺. Since the experimental setup does not allow the neutral production and the relative abundance of NeKr₂ is not known, I have some doubt about the relative cross section of ETMD.

I have a lot of doubt if the authors did well observed ETMD and I did not find convincing proofs.

Reviewer #2 (Remarks to the Author):

A - summary of the key results: The manuscript 'Charge transfer to ground state ions produces free electrons' investigates a cascade of autoionization processes taking place after the production of Ne²⁺ ions in NeAr clusters by an Auger decay. In the course of this cascade the initially produced Ne²⁺ ions are (partially) neutralized to Ne⁺ by electron transfer mediated decay (ETMD).

B - novelty & interest: This work is the first experimental evidence of cascades involving intramolecular Coulombic decay (ICD) and electron transfer mediated decay (ETMD). Until the present date only a theoretical prediction for these cascades existed on which this work is based upon. I expect these results to have impact on the field of autoionization processes and energy transfer in weakly bound systems since the results showcase that different de-excitation channels can be disentangled experimentally.

C - methodology: The experimental results also show very impressively the strength of the COLTRIMS setup, which is able to detect all charged entities arising from a process. In this particular case a COLTRIMS spectrometer seems to be the ideal setup to investigate a cascade of non-radiative decay mechanisms. Taking into account the high demands on the detection (ion-ion-ion-electron coincidence), the shown results are of high quality.

D, E, F - statistics, explanation of the method, improvements: Despite the importance of the results presented in the manuscript, it needs to be improved in some regards - especially concerning the presentation and description of the analysis of the data. These concerns regard figure 3 in particular:

- The most important concern is the spectral feature between 11 and 20 eV kinetic energy in figure 3g - spectral features in this region appear not to be significant considering the error bars and compared to the noise level at higher kinetic energy. This concern is amplified by the fact that the

respective spectra in figure 3g arise from spectral subtraction. In the case of the spectrum recorded at 888 eV photon energy, spectral subtraction has even been applied twice to the spectrum.

- About the subtraction of the Kr photoionization signal giving rise to figures 3d and 3e:
 - oA spectrum recorded at 860 eV photon energy has been subtracted from spectra recorded at 878 eV and 888 eV photon energy, respectively. Have the authors applied a correction factor to take into account the drop of the photoelectron cross section? If not - for what reasons / why is this drop negligible?

- oFor the sake of reproducibility it would be good to show the relative scaling of the spectra 3a/b and 3c before subtraction (including the region between 25 and 40 eV kinetic energy). This may be done in the supplementary information.

- In my opinion it is important to show the fitting of the spectra in figure 3f- especially with regard to the 'background signal' which contains the spectral features the authors discuss / show in figure 3g.

- The layout of the figures 3f and 3g can be much improved. In the current form it is not easy to distinguish between the two spectra. This may be improved e.g. simply by connecting the data points with lines to guide the eye.

Many of the above comments can be addressed in the supplementary information.

Some remarks on figure 1:

- According to the figure I would assume the lines symbolizing atomic states are shown on a vertical energy scale. If that is the case, why are the lines bent in the figure?

- Sometimes the number of (shown) electrons in the respective states is 1 and sometimes 2. Since only noble gas atoms are considered here, I would always expect (at least) 2 electrons / holes per state.

- The size of the symbols for the electrons / holes in the figure should be enlarged for the sake of visibility - there is space enough.

- The label for the states in the Kr atoms is missing.

- In 'step 2' - what is the state directly below the Ne 2p level in this figure? Why is it (apparently) unoccupied? Why is it in the figure when it is not involved in the decay at all?

Remarks on passages in the text:

- In the abstract and the introduction of the manuscript the authors create the impression that the Auger decay is always the typical decay channel. I advise to limit this statement to 'light atoms'.

- The introduction of the paper hinges on the difference in production of 'high energy' electrons and 'low energy' electrons. It would be convenient for readers not from the field if the authors state more precisely of which order of magnitude the energy difference is.

- Page 4, abstract at the top: 'It is expected that similar neutralization by ETMD is a dominant step of damaging, Auger driven decay cascades in an environment.' It is unclear what exactly the authors mean and only vaguely connected to the previous sentence in the text.

- The processes discussed in this manuscript take place on comparatively short timescales.

Assuming that the results are interesting for a broader audience, the authors should give a number for the relevant timescales at some point in the paper. This means essentially citing ref. 5.

- In the methods section the authors state that the branching ratio into the Ne²⁺ (2s-12p-1 1P) state is 17.2% without either giving an explanation where this number derives from or citing a source.

My concerns regarding the significance of one part of the data does not affect the quintessence of the paper since the spectral features in question are not related to the ETMD process, which is responsible for the (partial) neutralization of the Ne²⁺ ions.

G - previous work: The experiments presented in the manuscript base essentially on a theoretical prediction by Stumpf et al. (ref. 5). This is clearly stated in the text. Additional information, necessary for the understanding of branching ratios into different states bases on another reference given by the authors.

H - lucidity of the text / general: The manuscript is clearly written and brief (for the most part; see above remarks). The authors do not try to oversell their results and sketch the possible implications of their work including a brief outlook for future experiments. The abstract covers the essential of the manuscript. Thus, I recommend publication after minor revision.

Reviewer #3 (Remarks to the Author):

The paper by You et al. describes first experimental evidence for electron transfer mediated decay (ETMD) of core hole vacancies. After creation of a core hole, a Ne⁺ decays into dicationic states by Auger electron emission. Krypton atoms in the vicinity neutralize one of the 2p holes in Ne(2+) under emission of a low energy electron in the range between 0 and 10 eV.

The authors performed systematic studies irradiating mixed Ne - Kr clusters around the Ne K edge at a tunable soft x-ray synchrotron beamline. The process of ETMD is selected by logging coincidences of singly charged Ne together with a Kr atom pair. Electron kinetic energies are determined in coincidence with the 3 particle detection. Subtracting a background solely stemming from Kr ionization (by determining the electron kinetic energy exciting below the Ne 1s IP) and by subtracting the photoelectron peak, the authors identify a broad electron distribution ranging from 0 to about 10 eV kinetic energy. They assign the distribution to different ICD and ETMD channels. I think this paper is worth publishing in Nature Communications, it presents an original and relevant study. The process shown here is relevant because the low energy electrons are responsible for DNA strand breaking via dissociative attachment. I would therefore expect, that this science appeals to a larger scientific community, not only to atomic and cluster physicists. Of course, in nature the low energy electrons will be emitted from water or larger organic molecules. However, I really find it remarkable that the ion-electron coincidence methods allow to detect this signal under a background of all kinds of electrons produced in the soft x-ray interaction. The analysis and interpretation of the coincidence data is robust, the error has been treated with great care.

The authors arrive at the conclusion, that ETMD is really observed, and this is supported by calculations on the ratio of different decay channels. I guess the results are robust and reliable. The study beautifully shows the power of coincidence techniques and its applicability to new and relevant processes.

I have a few minor comments for the authors to consider:

- 1) What is the role of inelastic electron - atom collisions in the cluster? This will change the kinetic energy distribution. It is not mentioned yet, but I would assume that it is not too hard to give an estimate on this, taking into account that the size distribution of the clusters is known.
- 2) "We assume that branching ratios for Auger decay in atoms apply to our weakly bonded clusters as well. " - It would help to explain the reason behind this in a few sentences.
- 3) "In the larger mixed clusters measured here, the electron energies may, of course, shift somewhat. " - Could you estimate this 'somewhat'?
- 4) Is there a particular reason for choosing circular polarized excitation light?

The text is written clear, apart from the minor comments above. References are appropriate.

In the following, we give a point-by-point reply to all reviews we have received. We give our replies to reviewers #2 and #3 first, as some of our replies to reviewer #1 are implied (R: Reviewer, A: Answer, M: Modified text).

Reply to the reviewer #2

Reviewer:

D, E, F - statistics, explanation of the method, improvements:

Despite the importance of the results presented in the manuscript, it needs to be improved in some regards - especially concerning the presentation and description of the analysis of the data. These concerns regard figure 3 in particular:

- The most important concern is the spectral feature between 11 and 20 eV kinetic energy in figure 3g - spectral features in this region appear not to be significant considering the error bars and compared to the noise level at higher kinetic energy. This concern is amplified by the fact that the respective spectra in figure 3g arise from spectral subtraction.

Answer:

We thank the reviewer for recognizing the importance of our results. It is correct that our data, in particular for the high kinetic energy spectra, are influenced by noise due to imperfect statistics. This again is a result of the multi-coincidence nature of the ETMD signature we have strived to acquire, and can be improved in future experiments. Nevertheless, our result is statistically very significant. The error bars shown in Figure 3g result from a rigorous error propagation, including the additional error due to subtraction of the 860 eV background spectrum. (No additional error has been assigned to the subtraction of the Ne 1s photoline, as this is model-based and not subject to a statistical noise.) If we integrate over the 11–20 eV interval we arrive at signals of 383 (77) counts for the 878 eV spectrum and at 345 (90) counts for 888 eV, where round brackets give the standard deviation. (In our Figure 3g, ‘arbitrary units’ were introduced because spectra were scaled down with a factor near, but not identical to unity to match with the background spectrum, see below. Numbers given here are actual counts.) As we explain below, these numbers are a conservative lower limit for the ETMD signal. If we follow the advice of Reviewer #2 for the background subtraction procedure (see next comment), even larger numbers are received.

R: In the case of the spectrum recorded at 888 eV photon energy, spectral subtraction has even been applied twice to the spectrum.

A: Unfortunately it is necessary to subtract the Ne 1s photoelectrons to reveal the ETMD contribution to the spectrum. We are confident though in our representation of this feature by a Gaussian line, and do not believe that the second subtraction step is adding uncertainty to our interpretation.

R: •About the subtraction of the Kr photoionization signal giving rise to figures 3d and 3e:

o A spectrum recorded at 860 eV photon energy has been subtracted from spectra recorded at 878 eV and 888 eV photon energy, respectively. Have the authors applied a correction factor to take into account the drop of the photoelectron cross section? If not - for what reasons / why is this drop negligible?

A: As explained in the original manuscript, before background subtraction we have scaled our spectra acquired at 878 and 888 eV such that they match the intensity of the background spectrum in the featureless region above 20 eV kinetic energy, amounting to a multiplication by 0.90 and 0.82, resp. This procedure is independent of the energy dependence of the Kr cross section, and was chosen to also summarize all other factors for count rate differences between signal and background spectra. The comment of the reviewer is very legitimate though. We have therefore redone the analysis procedure by scaling all spectra according to the photon energy dependence of Kr cross section and the beamline flux, and to time-integrated sample pressure. The results of this exercise are detailed in Supplementary Notes 3 and 4, Supplementary Figs. 6 and 7 and Supplementary Tables 3 and 4 of the Supplementary Information and contrasted to our original analysis (Supplementary Figs. 4 and 5 and Supplementary Tables 1 and 2). It is seen that the latter procedure leads to a larger ETMD signal (less background is subtracted), and in some instances there is evidence that it is the more adequate way of data treatment. We nevertheless decided to keep the results of the original procedure in the letter text, but we are convinced now that thus we give a conservative lower limit for the amount of ETMD electrons produced. We thank the reviewer for prompting us in that direction.

R: o For the sake of reproducibility it would be good to show the relative scaling of the spectra 3a/b and 3c before subtraction (including the region between 25 and 40 eV kinetic energy). This may be done in the supplementary information.

A: We have added the Figure to the supplementary information as Supplementary Fig. 3.

R: • In my opinion it is important to show the fitting of the spectra in figure 3f- especially with regard to the 'background signal' which contains the spectral features the authors discuss / show in figure 3g.

A: We have added the fit profiles that were used to subtract the Ne 1s component to Figure 3f.

R: •The layout of the figures 3f and 3g can be much improved. In the current form it is not easy to distinguish between the two spectra. This may be improved e.g. simply by connecting the data points with lines to guide the eye.

A: We agree, and have connected the data points to make it easier to disentangle the two spectra. We have also changed colors of the data points to make it easier to distinguish them on a gray-scale print.

R: Some remarks on figure 1:

- According to the figure I would assume the lines symbolizing atomic states are shown on a vertical energy scale. If that is the case, why are the lines bent in the figure?
- Sometimes the number of (shown) electrons in the respective states is 1 and sometimes 2. Since only noble gas atoms are considered here, I would always expect (at least) 2 electrons / holes per state.
- The size of the symbols for the electrons / holes in the figure should be enlarged for the sake of visibility - there is space enough.
- The label for the states in the Kr atoms is missing.
- In 'step 2' - what is the state directly below the Ne 2p level in this figure? Why is it (apparently) unoccupied? Why is it in the figure when it is not involved in the decay at all?

R: We thank the reviewer #2 for these comments. We modified figure 1 according to the reviewer's suggestions.

R: Remarks on passages in the text:

•In the abstract and the introduction of the manuscript the authors create the impression that the Auger decay is always the typical decay channel. I advise to limit this statement to 'light atoms'.

A: We have modified the corresponding sentence in the introduction. We believe the statement in the abstract is fair, as 'typically' does not rule out that other decay channels may take place as well

Modified text is given with the next reply.

R: •The introduction of the paper hinges on the difference in production of 'high energy' electrons and 'low energy' electrons. It would be convenient for readers not from the field if the authors state more precisely of which order of magnitude the energy difference is.

A: We added energies to make order of magnitude of the energy difference clear.

M: The 1st line of Introduction section of page 2.

When a light atom is irradiated by X-rays, the most likely process is photoionization of an inner-shell electron, followed by the emission of one or several Auger electrons of relatively high energy ($>200\text{eV}$)¹.

When condensed matter is irradiated by X-rays however, the majority of electrons emitted into vacuum is of low energy ($<20\text{ eV}$).

R: •Page 4, abstract at the top: 'It is expected that similar neutralization by ETMD is a dominant step of damaging, Auger driven decay cascades in an environment.' It is unclear what exactly the authors mean and only vaguely connected to the previous sentence in the text.

A: We replaced the sentence with the following.

M: The 4th line of the 1st paragraph of page 4.

Such ETMD driven neutralization has a broad significance and was predicted to play an important role in radiation damage to biomolecules subjected to the action of ionizing radiation¹⁸.

R: •The processes discussed in this manuscript take place on comparatively short timescales. Assuming that the results are interesting for a broader audience, the authors should give a number for the relevant timescales at some point in the paper. This means essentially citing ref. 5.

A: The calculations in Ref. 5 (Ref. 14 in the new manuscript) show that ETMD lifetimes in NeKr_2 at the equilibrium nuclear configuration lie between 8 to 30 ps for different electronic states involved. Taking nuclear dynamics into account leads to the shortening of the lifetimes by about an order of magnitude. In a large mixed cluster, a Ne atom at the Ne-Kr interface has three nearest Kr neighbours. This should shorten the lifetimes by at least the factor of 3. The next but nearest neighbours will also lead to the shortening of the lifetimes, although not that strong. Finally, assuming similar effect from the nuclear dynamics as in the trimer case we obtain the lifetimes between a few hundred femtoseconds and one picosecond.

M: Towards the end of the introduction, page 4, sentence added.

According to Ref. 14, the process takes place within a few ps in a trimer, which could shorten to below

one ps due to environmental effects in larger systems.

R: •In the methods section the authors state that the branching ratio into the Ne²⁺ (2s-12p-1 1P) state is 17.2% without either giving an explanation where this number derives from or citing a source.

A: We added a reference number 25 to the corresponding sentence.

R: My concerns regarding the significance of one part of the data does not affect the quintessence of the paper since the spectral features in question are not related to the ETMD process, which is responsible for the (partial) neutralization of the Ne²⁺ ions.

A: We hope the reviewer #2 is now relieved from her/his concerns.

Reply to the reviewer #3

We are grateful to this reviewer who also considers our work is worth publishing in Nature Communications.

Reviewer: I have a few minor comments for the authors to consider:

1) What is the role of inelastic electron - atom collisions in the cluster? This will change the kinetic energy distribution. It is not mentioned yet, but I would assume that it is not too hard to give an estimate on this, taking into account that the size distribution of the clusters is known.

Answer: Briefly, due to the smallness of the clusters we measured, we estimate the influence of inelastic scattering to be in the few % at most (below 3 %). A somewhat extended discussion of inelastic channels and their potential influence is given in the Supplementary Note 5.

R: 2) "We assume that branching ratios for Auger decay in atoms apply to our weakly bonded clusters as well." - It would help to explain the reason behind this in a few sentences.

A: The major reason in our opinion is that in the van der Waals clusters composed of 'lighter' elements the binding is too weak and interatomic distances are too large to influence the Auger decay channel. Therefore, neither ICD-like amplitudes to core-level decay are expected (K. Kreidi et al., Phys. Rev. A **78**, 043422 (2008)), nor is there a chemical shift in the partial Auger rates (A. Lindblad et al., J. Chem. Phys. **123**, 211101 (2005)). We are not aware of a dedicated theoretical investigation of this fact. It seems to be a common wisdom though: see e.g. Kimura et al. Phys. Rev. A **87**, 043414 (2013). We now cite the Kreidi et al. paper in conjunction with the respective statement in the manuscript.

R: 3) "In the larger mixed clusters measured here, the electron energies may, of course, shift somewhat." - Could you estimate this 'somewhat'?

A: We can estimate these shifts by considering the effect of the cluster environment on the energy of different ions in the initial and final states of ICD and ETMD. In ICD between Ne²⁺ and Kr the charge on Ne as also its energy stabilization by the environment remain the same and do not contribute to the shift in the energies of ICD electrons. The shift due to the outer-valence ionization of Kr can be estimated from the photoelectron spectra in Kr pure clusters (Feifel et al., Eur. Phys. J. D **30**, 343 (2004)) and amounts to

0.7–1.1 eV relative to the atomic line depending on the state's multiplet. Therefore, the ICD electrons should be faster by about 0.7–1.1 eV. Since, the estimate was derived from the bulk values of Kr, the actual shift in the electron energies on the Ne-Kr interface might be less than this estimate. In ETMD, two singly ionized Kr atoms appear, while Ne charge goes from 2+ to 1+. We estimated the stabilization energy of Ne²⁺ on the Ne-Kr interface from the pair binding energies in Ne-Ne and Ne-Kr dimers. We assumed the fcc structure in both Ne layers and Kr core. The Ne²⁺ has 9 nearest Ne neighbors. It also has 3 nearest, 3 next nearest and 6 next-next nearest Kr neighbors. Taking into account these neighbors alone we estimated the stabilization energy of Ne²⁺ as 2.97 eV. For Ne⁺ assuming 9 nearest Ne neighbors, 1 nearest, 3 next nearest and 6 next-next nearest Kr neighbors we obtained 0.82 eV. Together with the energy stabilization of two Kr⁺ amounting to 1.4–2.2 eV we obtain that the final state is stabilized and the ETMD electrons are shifted by –0.75 to 0.05 eV relative to the NeKr₂ trimer.

R: 4) Is there a particular reason for choosing circular polarized excitation light?

A: We used circular polarization because we can get the highest flux for the incident light in the beamline. We modified a sentence as the following.

M: The 3rd line of Method section on page 9.

Circular polarization of the incident light was chosen in order to maximise the photon flux at the used photon energies (860, 878 and 888 eV)²³.

Reply to reviewer #1

We thank this reviewer for her/his through reading of our manuscript and her/his comments to improve it.

Reviewer: The authors were able to extract coincidences between three ions : Ne⁺ and two Kr⁺ ions. From the experimental point of view, it would have been much better to learn about the Kinetic Energy Released of the dissociative NeKr₂ triply charged ion. Does the KER correspond to what can be expected from a simple Coulomb explosion picture ?

A: We thank the reviewer for bringing up this important point. In figure R1, we plot the number of events vs. total ion energy (after subtraction of contributions from Kr photoionization). If we assume the equilibrium geometry of the NeKr₂ trimer reported by Stumpf et al. (Phys. Rev. Lett. **110**, 258302 (2013)), the sum of the energies of three ions (Ne⁺ and two Kr⁺) is 10.08 eV, which is in reasonable agreement with the high kinetic energy flank of the peak in the spectrum. The plot extends towards lower total ion energies ('kinetic energy release', KER) and the peak is broad, because our clusters were on average larger than a trimer and neutrals fragments take some momentum. The above discussion has been described in Supplementary Note 2 and the ion intensities vs. total ion energy before and after the subtraction have been shown in Supplementary Figs. 1 and 2, respectively.

Figure R1. Sum of the energies of the three ions.

M: We have added the following text towards the end of the Discussion section.

Our interpretation is additionally supported by an analysis of the ion kinetic energy release (KER), shown in the Supplementary Information. The KER spectrum clearly shows a component related to Ne photoionization, the energies of which are in reasonable agreement with a Coulomb explosion of Ne^+ and 2Kr^+ , in which some momentum is taken away by neutral fragments.

R: The data treatment (subtraction of below Neon 1s threshold) and comparison with above Neon threshold measurements suffers from the photoelectron peak which stands up in the middle of the spectrum and with low statistics. It would be nicer to appreciate the number of events measured to appreciate the statistical uncertainty.

A: We have added the number of events to the caption of figure 3.

M: In caption of Fig. 3

In total, in panels (a), (b) and (c) 18616, 17610 and 13086 events are plotted, respectively.

R: The figure 3c (measurements below the Ne 1s threshold) exhibits almost the same 2D spectrum when measured 8 or 18 eV above the Ne 1s threshold. It is then hard to believe that the authors well measured ETMD above the threshold.

A: Nevertheless, the differences in the spectra we observe, are statistically clearly significant. In order to support this statement, we prepared Supplementary Figure 3 which shows electron spectra in coincidence with Ne^+ and two Kr^+ for energy region between 0 and 40 eV before subtraction. We normalized the spectra so that integrated intensities of 878 eV and 888 eV spectra between 25 and 40 eV fit to that of 860 eV spectrum. Then, we found a significant difference in the lower kinetic energy region, which comes *in addition* to the photoelectron peaks. The statistical significance of our experimental result is explained in detail also in our first reply to Reviewer #2, and we refer to the discussion given there.

R: I also have some doubt about the relative cross section of the ETMD. The authors claim that 70% of the 1s ionization event leads to ETMD. I wonder if the authors did take into account some events leading to, for instance $\text{Ne}^+ + \text{Kr}^+ + \text{neutral Kr}$ or to Ne^{++} . Since the experimental setup does not allow the neutral production and the relative abundance of NeKr_2 is not known, I have some doubt about the relative cross

section of ETMD.

A: The reviewer is correct in bringing up this point, since the statement in our original manuscript was imprecisely formulated, such that it read more general than intended by the authors. ETMD after Auger decay is an interface phenomenon, and what we would like to say is that at least 70 % of all Ne states core ionized *at the NeKr interface* undergo subsequent ETMD. This, we believe, has clearly been shown by the account of decay channels in Table 1, and by further arguments e.g. on background subtraction presented now in the Supplementary Information. If we ask which decay modes of a Ne core hole state could produce a completely different signature in the fragment spectra, the only alternative that comes to our minds is Auger decay, followed by radiative charge transfer (RCT) to a neighbouring Ne atom. Indeed, this process would lead to the production of only two charged fragments (2Ne^+), and would thus be exempt from our analysis. Based on our estimates of the RCT rate and on experimental results on this process e.g. in Ar clusters, we believe it is highly unlikely that this channel plays a role as soon as ETMD is available as a decay channel. (In Ar clusters for example, RCT is only observed from states for which all radiationless decay channels are closed.) Radiative Charge Transfer might play a role however for clusters in which only a single Kr atom is present. We have not looked for this decay process in the analysis presented here.

In order to fairly inform our readers of this caveat, the above discussion, with adequate references, has been described in Supplementary Note 1. We have changed the manuscript text to:

M: Towards the end of subsection “Electron energy spectra”.

This implies that when a Ne 1s photoionization event at the Ne-Kr interface occurs, almost always an electron with a low kinetic energy of several eV is emitted from the Ne-Kr mixed cluster.

R: I have a lot of doubt if the authors did well observed ETMD and I did not find convincing proofs.

A: Critical thinking is an important part of the peer review process, and we appreciate it. We hope to have convinced also reviewer #1 of the soundness of our findings.

Apart from these changes in order to accommodate criticism from the reviewers, we have included a sentence on the effect of double Auger decay, which is a minor channel and had been neglected when writing the initial version of the manuscript. Meanwhile we have started work on refined theoretical estimates taking this channel into account. This does not lead to differences in the results between which the current experimental data could discriminate, though. Moreover, very complex decay cascades become possible then, which will be the subject of a future publication. We think it is fair though to say:

M: End of first par. in ‘Discussion’

Qualitatively similar, but more involved decay cascades are also expected after double Auger decay to Ne^{3+} (6 % ²¹ branching ratio).

We changed styles of text and figures and renumbered references following the “MANUSCRIPT CHECKLIST”. Ref. 21 was replaced by the latest literature. Some typos and mistakes in writing were corrected. Changes are highlighted by yellow patterns in “Related Manuscript File”.

REVIEWERS' COMMENTS:

Reviewer #1 (Remarks to the Author):

The authors are dealing with a very important and interesting problem. The authors did well answer to the questions. Nevertheless, even with these more detailed explanations on the statistical errors, I'm not at all convinced that the authors did prove the occurrence of the mentioned ETMD process. I encourage them to spend more acquisition time on this issue and to acquire the same kind of spectrum at much higher photon energy where the Neon photoelectron peak would not be the dominant peak in the middle of the energy window of interest. I do not recommend to publish this manuscript.

Reviewer #2 (Remarks to the Author):

I thank the authors for their accurate reply and I am fully satisfied with the answers / the changes in the manuscript. My concerns concerning the statistical significance are resolved - to a large degree due to the extensive supplementary information. I appreciate the effort the authors have invested in writing the SI and describing their methods clearly.

Reviewer #3 (Remarks to the Author):

I would like to thank the authors for carefully considering all my remarks and questions. In short, I am happy with the changes. The explanations in the supplement and additions to the paper certainly increase the readability and understanding of that paper. If the other referees agree for their part, I will not stand in the way of publishing.

In detail:

1) What is the role of inelastic electron – atom collisions in the cluster? This will change the kinetic energy distribution. It is not mentioned yet, but I would assume that it is not too hard to give an estimate on this, taking into account that the size distribution of the clusters is known.

I think this critique has been appropriately addressed in Supplement 5. The authors consider two channels, 1) Losses of Ne 1s photoelectrons and high kinetic energy ETMD electrons by inelastic scattering. 2) Production of additional low kinetic energy electrons by inelastic scattering of Auger electrons, or of Ne valence shell or Kr photoelectrons. The cross sections for these channels are estimated by comparisons to literature and it seems that the 6% upper cross section is below the experimental error.

2) "We assume that branching ratios for Auger decay in atoms apply to our weakly bonded clusters as well." – It would help to explain the reason behind this in a few sentences.

The question is appropriately answered and a paper 'Kimura et al. Phys. Rev. A 87, 043414 (2013)' is cited to strengthen this point.

3) "In the larger mixed clusters measured here, the electron energies may, of course, shift somewhat." - Could you estimate this 'somewhat'?

The question is answered using charge state and geometry of the nearest neighbour environment around the emission site.

4) Is there a particular reason for choosing circular polarized excitation light?

It was pointed out that circular polarization delivers the highest flux at the particular beamline

used. There does not seem to be any deep physical reasoning behind that and an appropriate remark is added to the text.

Reply to the reviewer #1

Reviewer:

The authors are dealing with a very important and interesting problem. The authors did well answer to the questions. Nevertheless, even with these more detailed explanations on the statistical errors, I'm not at all convinced that the authors did prove the occurrence of the mentioned ETMD process. I encourage them to spend more acquisition time on this issue and to acquire the same kind of spectrum at much higher photon energy where the Neon photoelectron peak would not be the dominant peak in the middle of the energy window of interest.

I do not recommend to publish this manuscript.

Answer:

Indeed the statistics of our experiment is limited, as this is a complicated multi-coincidence experiment. Nevertheless we think that our data rigorously establish the existence of Auger-ETMD cascade processes, although they may not be sufficient to assess quantitative details. This view is now shared by several other reviewers. We have added a description about this limitation to the text.

Added text: The 2nd paragraph of page 9.

While our experiment has provided firm evidence for the existence and the efficiency of Auger-ETMD cascade processes, the statistics of our data was not sufficient to assess it in all quantitative detail. More sophisticated modeling of the cascade processes as well as extended measurements should further improve our understanding.

Reply to the reviewer #2

Reviewer:

I thank the authors for their accurate reply and I am fully satisfied with the answers / the changes in the manuscript. My concerns concerning the statistical significance are resolved - to a large degree due to the extensive supplementary information. I appreciate the effort the authors have invested in writing the SI and describing their methods clearly.

Answer:

We thank the reviewer #2 who fully agree our revisions and suggest to publish our manuscript in Nature Communications.

Reply to the reviewer #3

Reviewer:

I would like to thank the authors for carefully considering all my remarks and questions. In short, I am happy with the changes. The explanations in the supplement and additions to the paper certainly increase the readability and understanding of that paper. If the other referees agree for their part, I will not stand in the way of publishing.

In detail:

1) What is the role of inelastic electron – atom collisions in the cluster? This will change the kinetic energy distribution. It is not mentioned yet, but I would assume that it is not too hard to give an estimate on this, taking into account that the size distribution of the clusters is known.

I think this critique has been appropriately addressed in Supplement 5. The authors consider two channels, 1) Losses of Ne 1s photoelectrons and high kinetic energy ETMD electrons by inelastic scattering. 2) Production of additional low kinetic energy electrons by inelastic scattering of Auger electrons, or of Ne valence shell or Kr photoelectrons. The cross sections for these channels are estimated by comparisons to literature and it seems that the 6% upper cross section is below the experimental error.

2) “We assume that branching ratios for Auger decay in atoms apply to our weakly bonded clusters as well. “ – It would help to explain the reason behind this in a few sentences.

The question is appropriately answered and a paper ‘ Kimura et al. Phys. Rev. A 87, 043414 (2013)’ is cited to strengthen this point.

3) “In the larger mixed clusters measured here, the electron energies may, of course, shift somewhat. “ - Could you estimate this ‘somewhat’?

The question is answered using charge state and geometry of the nearest neighbor environment around the emission site.

4) Is there a particular reason for choosing circular polarized excitation light?

It was pointed out that circular polarization delivers the highest flux at the particular beamline used. There does not seem to be any deep physical reasoning behind that and an appropriate remark is added to the text.

Answer:

We are grateful the reviewer #3 who fully agree our revisions and suggest to publish our manuscript in Nature Communications.